# Molecular Pathways for Muscle and Adipose Tissue Are Altered between Beef Steers Classed as Choice or Standard

**DOI:** 10.3390/ani13121947

**Published:** 2023-06-10

**Authors:** Sarah A. Haderlie, Jordan K. Hieber, Jane A. Boles, James G. Berardinelli, Jennifer M. Thomson

**Affiliations:** Department of Animal and Range Sciences, Montana State University, Bozeman, MT 59717, USA

**Keywords:** carcass quality, beef cattle, adipose tissue, muscle tissue, transcriptomics

## Abstract

**Simple Summary:**

The eating experience of beef is strongly influenced by the amount of marbling or intramuscular fat. Beef steers are fed to a desired level of fatness in order to maximize the probability of a positive eating experience. This work aims to study the genes expressed in muscle and adipose tissue as beef steers deposit body fat to better understand the regulation of this important process and to more effectively predict positive eating experiences. This research identified genes that are altered as beef animals deposit fat in both the muscle and adipose tissues and the processes that are being altered, including the metabolism of fat and energy. The work also identified genes related to inflammation and injury in the adipose tissue that warrant additional research to further understand their role in beef animals.

**Abstract:**

Targets for finished livestock are often determined by expected fat, either subcutaneous or intramuscular. These targets are used frequently to improve eating quality. Lower intramuscular fat, lack of product uniformity, and insufficient tenderness can negatively impact beef acceptability. This study aimed to investigate the differences in gene expression that alter metabolism and intercellular signaling in the muscle and adipose tissue in beef carcasses at different fat endpoints. In this study, longissimus thoracis muscle samples and adipose tissue were collected at harvest, and RNA was extracted and then sequenced using RNAseq. Differential expression was determined using edgeR, and *p*-values were adjusted using the Benjamini–Hochberg method. A corrected *p*-value of 0.005 and log_2_ (fold change) of >1 were the threshold to identify differential expression. Comparison between intermuscular and subcutaneous fat showed no differences in the genes activated in the two adipose tissue depots, suggesting that subcutaneous fat was an adequate sample. Carcass data allowed the classification of carcasses by USDA quality grades (marbling targets). In comparing muscle from Standard and Choice carcasses, 15 genes were downregulated, and 20 were upregulated. There were 49 downregulated and 113 upregulated genes comparing adipose tissue from Standard and Choice carcasses. These genes are related to the metabolism of fat and energy. This indicates that muscle transcript expression varies less than adipose. In addition, subcutaneous fat can be used to evaluate transcript changes in fat. However, it is unclear whether these fat tissues can be used as surrogates for marbling.

## 1. Introduction

Intramuscular fat has been used for quality assurance systems within the US and Australia to improve the consistency of tenderness and improved eating experiences [1]. Iida et al. [2] reported that sensory scores for juiciness and tenderness increased as the fat level increased in steaks from Japanese Black steers. Other researchers have reported that increasing levels of marbling results in a higher likelihood of a tender product [3,4]. In general, as subcutaneous fat increases, marbling also increases, and producers have used visual observations of fat and live body weight to determine harvest times. As an animal matures, it exhibits significant shifts in physiological responses. For example, as muscle growth approaches its peak, energy utilization for muscle growth is reduced, resulting in increased energy storage [3,5]. These changes encourage the deposition of fat, including marbling. However, animal-to-animal variation results in differing levels of marbling and subcutaneous fat. Thus, there is a need to identify what changes occur at the molecular level that initiate fat deposition to help predict final fat deposition.

While advances have been made in genetic selection and objective measures of fat, animals are still not reaching the desired body composition endpoints after similar days on feed. Engle et al. [6] reported a significant number of differentially expressed genes in longissimus lumborum muscle from Standard and Choice carcasses (1257 genes, *p* < 0.051) in Hereford cattle. Functional analysis of these genes revealed differences in the underlying pathways regulating muscle cell growth and proliferation. Biological processes of upregulated genes were associated with signaling pathways associated with inflammation, growth, and metabolism. Furthermore, the upregulation of processes associated with the extracellular matrix, stem cell differentiation, and focal adhesion was observed. Further investigation into the changes occurring during fat deposition to better predict and select animals that consistently achieve the desired level of marbling and fat in a predictable fashion is important. This would allow for more efficient feeding and management of finishing cattle. We hypothesize that gene expression in adipose tissue changes during fattening and that coordinated changes in expression between muscle and adipose tissues can be detected. This work aimed to provide new insights into the differences in muscle and fat tissue gene expression and metabolic pathways when carcasses have reached marbling associated with different USDA quality grades.

## 2. Materials and Methods

Data collection protocols were approved by the Montana State University Agriculture Animal Care and Use Committee (Protocol No. 2015-AA17).

Fifteen Angus-sired steers were selected at weaning based on weight and date of birth. Steers were moved to the Montana State University Bozeman Area Research and Teaching Farm and placed in a single pen in the feedlot. At the start of the study, steers weighed an average of 313 ± 14 kg. Each steer received a Synovex One Feedlot implant per standard feedlot protocol. They were fed an ad libitum diet of hay for two weeks to acclimate steers to pens and were then started on a 6-week, step-up program to build up to the full ration (Table 1), which was a diet of 75% shelled corn, 18% hay, and 7% finisher pellet, fed in a bunk daily and had free access to water. Steers were randomly allocated to one of three endpoints based on body weight and visual evaluation, with average endpoint weights of 431 kg, 522 kg, and 612 kg targeting marbling levels for Standard, Select, and Choice quality grades, respectively. Animals were on feed an average of 124 days with a standard deviation of 35 days. These weight-based endpoints achieved the desired marbling endpoints (Table 2). Animals were transported 60 miles to a small commercial processor and harvested following normal industry standards. Carcass data were collected 24 h postmortem to calculate yield grade and determine quality grade following USDA guidelines [7]. The carcasses were ribbed between the 12th and 13th rib, exposing the longissimus dorsi. Marbling determination was performed a minimum of 20 min after ribbing to give time for the muscle to oxygenate. Carcass characteristics were measured by an individual with 30 years of experience in carcass data collection. Carcass measurements were collected as outlined by Engle et al. [6].

Striploins were removed at 24 h postmortem and transported to the Montana State University Meat Laboratory (Bozeman, MT, USA). The striploins were cut into 2.54 cm steaks starting at the anterior end. Steaks were individually vacuum packaged. Steaks were randomly assigned to different days of postmortem aging (24 h, 3, 7, 14, or 21 d postmortem). One steak collected from approximately the 13th rib area of the loin was cut into five equal pieces for the myofibrillar fragmentation index (MFI). These pieces were vacuum packaged and randomly assigned to the different aging periods as described for steaks. After aging, vacuum-packaged meat was placed in a −20 °C freezer until analysis. Steaks were taken from a −20 °C freezer and placed in a 2 °C cooler approximately 24 h prior to cooking. Steaks were blotted, tagged, weighed, and a single copper constantan thermocouple (OMEGA Engineering, INC, P.O. Box 4047, Stamford, Connecticut) was placed in the center to track the endpoint temperature. They were then placed on an aluminum-covered broiler pan and placed under the broiler of a conventional oven 10.16 cm below the heating element. Steaks were then cooked on broil until reaching an internal temperature of 35 °C. At that point, the steaks were turned and cooked on the other side until reaching an internal temperature of 70 °C [8]. Steaks were cooled in a 2 °C cooler. After cooling for a minimum of 45 min, steaks were taken from the cooler, blotted with towels, and weighed. A minimum of five samples were taken parallel to the muscle fiber, resulting in square samples of 1.27 × 1.27 cm. The samples were then sheared using a TMS 30 Food Texturometer fitted with a Warner–Bratzler shear attachment. The average of the samples sheared was used for statistical analysis.

The MFI was determined following the procedures reported by Culler et al. [9], as modified by Hopkins et al. [10]. Two samples per steer per time point were averaged to yield five measurements per steer. The MFI average for each sample was calculated, and the average was used for statistical analyses (Table 3). Carcass data were analyzed using the Proc GLM procedure in SAS (v10.2) (SAS Institute, Cary, NC, USA) with quality grade class as an independent variable. MFI and shear force were analyzed using the Proc GLM procedure of SAS with quality grade and day of aging as independent variables. The interaction by day was tested, but, was not significant, so it was dropped from the analysis. The LSMEANS procedure was used to calculate the means and determine significance. Significance was set at a threshold of *p* < 0.05.

Subcutaneous and intermuscular adipose tissue samples were taken between the 4th and 5th rib at the time of harvest and homogenized immediately in Triazol and then transported on dry ice for subsequent RNA extraction. Longissimus thoracis muscle samples from the same area were snap frozen. All samples were collected within 30 min from the time of stun. Frozen muscle samples and homogenized adipose tissue sample RNA were extracted using a Qiagen RNAeasy Plus Universal Midi kit according to the manufacturer’s recommendations (Qiagen LLC, Georgetown, MD, USA). Briefly, this procedure involved manually homogenizing tissue in the Qiazol lysis reagent (Qiagen LLC, Georgetown, MD, USA). The lysate was centrifuged with chloroform and then the upper aqueous phase was mixed with ethanol and loaded into a spin column tube, where it binded the matrix. It was then washed and eluted off the spin column.

A total of 3 μg of RNA per sample was used to generate sequencing libraries using the NEBNext Ultra RNA Library Prep Kit^®^ from Illumina (San Diego, CA, USA) following the manufacturer’s instructions and index codes were added to each sample. RNA was enriched for mRNA, and cDNA libraries were created using the AMPure XP system (Beckman Coulter, Beverly, MA, USA). Libraries were sequenced on an Illumina Hiseq platform, and 125 bp/150 bp paired-end reads were generated. Read quality metrics are shown in Table 4.

The reads were aligned to reference genome *Bos taurus* UMD 3.1.1 [11]. The FPKM, the expected number of fragments per kilobase of transcript sequence per millions of base pairs sequenced for each gene, was then calculated based on the length of the gene and the reads count mapped to the said gene.

Prior to differential gene expression analysis, for each sequenced library, the read counts were adjusted by the edgeR program package [12] through one scaling normalized factor. Differential expression analysis of two conditions was performed using the DEGSeq R package (1.20.0) [13]. The *p*-values were adjusted using the Benjamini–Hochberg method. A corrected *p*-value of 0.005 and fold change >1 were set as the threshold for significantly differential expression.

Gene Ontology (GO) enrichment analysis of differentially expressed genes was implemented by the GOseq R package [14], in which gene length bias was corrected. GO terms with an FDR corrected *p*-value < 0.05 were considered significantly enriched by differentially expressed genes. KEGG [15], a database resource for understanding high-level functions and utilities of the biological system, was used, along with KOBAS software [16], to test the statistical enrichment of differentially expressed genes in KEGG pathways. Gene and pathway networks were generated through the use of IPA (QIAGEN Inc., Hilgen, Germay, https://www.qiagenbio-informatics.com/products/ingenuity-pathway-analysis (accessed on 27 January 2023)).

## 3. Results and Discussion

As expected, carcass weights from animals with Choice quality grade were greater (*p* = 0.002) than carcass weights from carcasses with less marbling that would result in Select or Standard quality grades (Table 2). This is not unexpected as the steers resulting in Choice carcasses were on feed longer than the steers of the other quality classifications (150 days versus 73 and 94). Furthermore, the fat thickness was significantly greater (*p* = 0.007) for the carcasses graded as Choice and Select than the carcasses graded as Standard. Other researchers have reported increased fat thickness as the carcass grade and carcass weight increase [17]. Conversely, the ribeye area was not different between the carcasses of different grades. This is in contrast to other reported results where the ribeye area was observed to be larger when carcass weights were higher [18,19,20]. Fat thickness was greater (*p* = 0.007) for the Choice and Select carcasses than for Standard carcasses (Table 2). This is expected, as increasing marbling is needed for higher-quality grades and is usually associated with a higher fat content in the whole carcass [3,6].

Shear force values for steaks from the Choice and Select carcasses were significantly lower than values for steaks from Standard carcasses (Table 3). Mixed results have been reported for differences in tenderness associated with different marbling degrees. Vierck and co-workers [4] reported differences in tenderness between steaks from Choice and Select carcasses, but no differences were observed between the different levels of Choice carcasses. However, other researchers have reported that marbling had little effect on tenderness [21,22,23]. Additional studies evaluating very high levels of intramuscular fat have consistently shown improved tenderness as the marbling increased [2,5]. Nishimura and colleagues [5] suggested changes in connective tissue structure (electron microscopy) as the fat level in the longissimus contributed to the increase in tenderness. This suggests that comparisons at higher levels of marbling may confound the information on how marbling affects tenderness.

The myofibrillar fragmentation index (MFI) was significantly lower for steaks from the Choice and Standard carcasses when compared to steaks from Select carcasses (Table 3). Higher MFI values are associated with a greater breakdown of fibers and have been related to improved tenderness. Kim and Lee [24], however, saw no effect of marbling on MFI when comparing steaks from Hanwoo cattle of varying grades. The differences in reported MFI values do not match the differences in shear force values. Researchers have reported that higher MFI values are correlated to lower Warner–Bratzler shear values [25]. The aging of steaks resulted in a decreased shear force for up to 14 days. The greatest change, however, was observed in the first 3 days of aging. Olson et al. [26] also reported the greatest change in shear force in the first 3 days of aging. Ilian and co-workers [27] reported similar results. However, Bratcher and co-workers [28] found improvement in shear force values up to 14 days of aging. MFI values increased up to 7 days of aging but were only significantly different between 1 day of aging and the rest of the aging times. Ilian et al. [27] reported significant increases occurred in MFI gradually during days 2 and 3 postmortem, but non-significant changes occurred after 7 days. These researchers also found strong correlations between the kinetics of tenderization and MFI. These data suggest that the sarcomeric structure’s dissolution rate is the fastest in the early postmortem period and slows as the time postmortem increases.

The RNAseq experiment yielded the expected number of transcripts with, on average, over 50 million mapped reads and high-quality data, as shown in Table 4.

In four cases, the intermuscular fat RNA was of insufficient quality or quantity for sequencing. We replaced these samples with subcutaneous adipose tissue. Figure 1 shows the Pearson correlations between each individual sample and the other samples included in the dataset. There are high correlations (>94% on average) between muscle samples, with one sample that was potentially contaminated by connective or vascular tissue, and a high correlation (>90%) between adipose samples, including those subcutaneous samples that were used to replace intermuscular samples. This shows that at these carcass endpoints, there are no detectable differences in gene expression between intermuscular and subcutaneous adipose tissues. In addition, there were more differentially expressed genes in adipose tissue (Table 5) than in muscle tissue (Table 6).

In the comparison between adipose tissue from the Standard and Select carcasses, 4 genes were upregulated, and 29 were downregulated (Table 5). Notably, two genes associated with WNT signaling were downregulated in adipose tissue from the Select carcasses. WNT signaling suppresses adipogenesis by blocking the activation of *PPARA* and *CEPBA*, which are the major regulators of adipogenesis [29]. In addition, the downregulation of WNT signaling is related to low insulin sensitivity [30]. One upregulated transcript is *BOLA-DMB,* which codes for MHC type 2 and indicates the presence of immune cells such as macrophages [31]. Furthermore, class 2 MHC genes are increased in obesity-induced adipose tissue inflammation [32]. In comparing adipose tissue from Select to Choice carcasses, 8 genes were downregulated, and 15 genes were upregulated (Table 5). Leptin and *ACC1* (Acetyl CoA Carboxylase 1) are two examples of downregulated genes, and this could lead to insulin resistance and potentially inhibit lipogenesis [33,34]. Reduced leptin expression has been shown in animal models of obesity-induced adipose tissue inflammation [35]. There were 49 downregulated genes and 113 upregulated genes in the comparison between adipose tissue from Standard and Choice carcasses (Table 5). Downregulated genes included the following: *CAB39L*, *FGF-1*, *GRIN1*, *LEP*, *HK2*, *YWHAG*, *ACC1*, *SCD1*, and *ELOVL3*. These genes are mostly related to fat and energy metabolism and promote adipogenesis and fat deposition (Table 7, Table 8 and Table 9) [3]. For another example, the downregulation of *ACC1*, *HK2*, and the upregulation of *EIF43BP1* would inhibit protein synthesis and increase lipogenesis. Furthermore, the downregulation of leptin can cause insulin resistance [32,33,34,35,36], and the upregulation of gluconeogenesis, which alters fat metabolism. Additionally, a number of immune markers were upregulated in the adipose tissue from Choice animals compared to Standard animals, which may be due to inflammation and immune activation in the adipose tissue. This pattern has been previously observed in related studies [32,33,34,35,36]. The GO terms enriched in the comparison of Select to Standard adipose tissue are lipid metabolic process and negative cholesterol and lipid storage regulation with primarily upregulated genes in these categories (Table 7). In comparing Select to Choice animals, adipose tissue development and lipid biosynthesis are downregulated in the enriched pathways, potentially indicating lipid deposition is slowing in the Choice animals (Table 8). In comparing Choice to Standard adipose tissue, we can observe the enrichment of upregulated GO terms related to adipose tissue development and lipid biosynthesis, which was expected as the fat deposition is much higher in the Choice carcasses (Table 9).

In comparing muscle from the Standard and Choice carcasses, 15 genes were downregulated, and 20 were upregulated (Table 6 and Table 10). The insulin receptor substrate 1 (*IRS 1*) gene was the only known functionally important gene to be differentially expressed. No transcripts associated with muscular hypertrophy were differentially expressed. This could be in relation to the lack of observed statistically significant differences in REA observed in this study. Our lab previously showed a number of differentially expressed genes in muscle tissue from carcasses of varying USDA quality grades [6]. Other labs have also reported differentially expressed genes in the muscle during fattening [37,38]. The lack of differentially expressed genes with known roles in meat tenderness is somewhat unexpected and indicates that the changes that occur in muscle that improve tenderness are not under transcriptional control during this time. In Table 10, it is shown that the only GO terms enriched were related to microtubule formation and organization in comparing muscle from the Choice and Standard carcasses.

Differentially expressed gene lists were uploaded to Ingenuity Pathway Analysis software (Qiagen Inc.), and network analysis was conducted. In comparing Choice to Standard adipose tissue, a network annotated as Endocrine system disorders, Metabolic Disorders, and Organismal injury was highlighted. This network is centered on insulin and growth hormones, which are not expressed in adipose tissue (Figure 2). Still, most of the molecules associated with these are upregulated in our dataset. This also indicates that inflammatory processes are being activated in the adipose tissue as fat deposition increases.

## 4. Conclusions

The data clearly suggest that subcutaneous and intermuscular adipose tissue have similar transcript profiles in the fat deposition period that was measured in this study. In addition, this work examined how the transcriptome profile changes as animals stop depositing protein and deposit lipids. These data indicate that nuclear or post-translational regulators potentially regulate muscle tissue upon reaching a mature size instead of transcriptional control. The transcriptomic profile of adipose tissue showed increased lipogenic, adipogenic activity and inflammatory processes. Additional research is needed to investigate the potential role of inflammatory activity in relation to fat deposition and carcass quality grade metrics.

## Figures and Tables

**Figure 1 animals-13-01947-f001:**
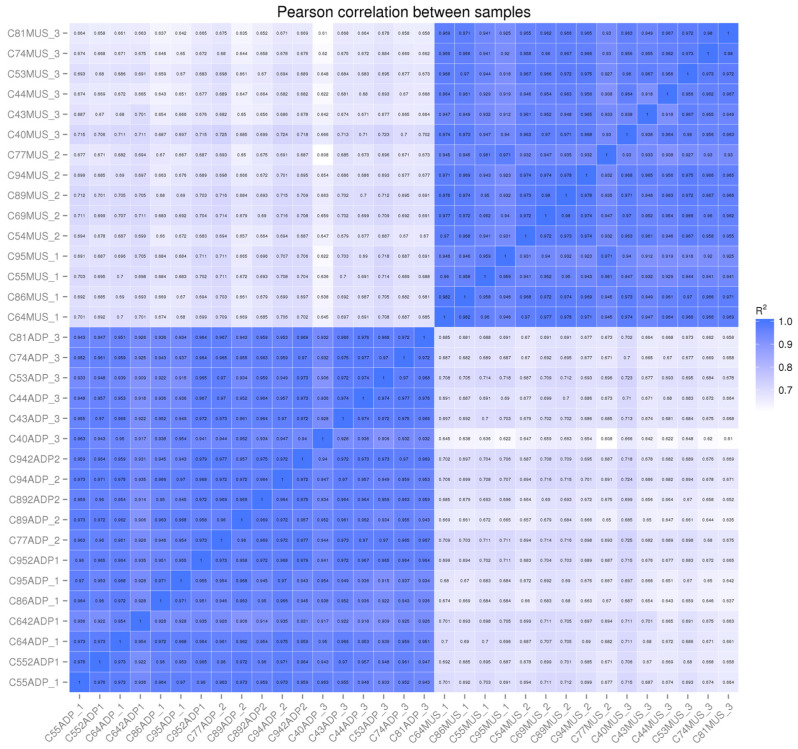
Pearson correlations between individual muscle and adipose sample FPKM value averaged across transcripts showing high correlations between samples of the same tissue and low correlation between samples of different tissues (muscle samples have an MUS appended to the animal number, and adipose samples have ADP appended to the animal number).

**Figure 2 animals-13-01947-f002:**
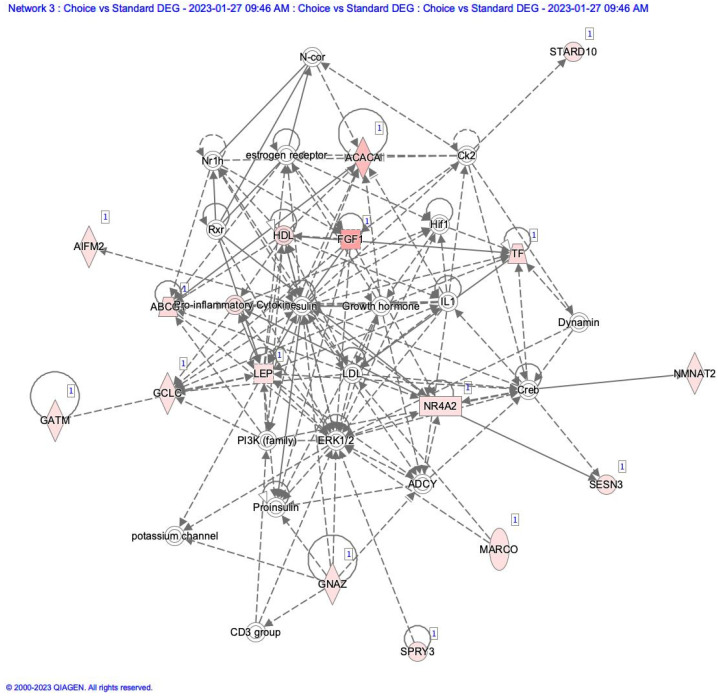
Network diagram of differentially expressed genes compared to adipose tissue from Choice to Standard quality grade samples. Items highlighted in red show higher numbers of elements in Choice than in Standard adipose samples.

**Table 1 animals-13-01947-t001:** Diet composition of feedlot diet fed to beef steers.

Item, % (DM Basis)	Diet Composition ^1^
Beef Finishing Pellet	6.3
Corn Dry Grain	75.18
Grass Hay	18.52
Nutrient Composition ^2^	
DM, %	86
CP, %	11.7
TDN, %	83.72
NDF, %	19.29
Ca, %	0.68
P, %	0.32
S, %	0.18
K, %	0.97
NEm, Mcal/kg	1.75
NEg, Mcal/kg	1.18

^1^ Diet consists of corn grain, grass hay, and Beef Finisher Pellet 40-20 R400 (CHS Nutrition, Great Falls MT). ^2^ Values based on proximate analysis of the individual ingredients.

**Table 2 animals-13-01947-t002:** Carcass characteristics of steaks from carcasses classified as Choice, Select, and Standard.

	Choice (n = 6)	Select (n = 5)	Standard (n = 4)
ADG (kg)	1.85	1.81	1.52
Carcass Wt. (kg)	339.3 ^a^	275.7 ^b^	243.1 ^b^
Fat Thickness (cm)	1.4 ^a^	1.0 ^a^	0.5 ^b^
Ribeye Area (cm^2^)	70.9	64.0	66.9
Marbling Score ^y^	510 ^a^	382 ^b^	285 ^c^

^a,b,c^ means values within a row with differing superscripts are significantly different (*p* ≤ 0.05); ^y^ marbling scores: 200 = traces, 300 = slight, 400 = small, 500 = modest, 600 = moderate. ADG: Average Daily Gain.

**Table 3 animals-13-01947-t003:** Effect of quality classification on the shear force and myofibrillar fragmentation index (MFI) of beef strip steaks.

Quality Classification ^1^	Shear Force (N) ^3^	MFI	Shear Force (N) ^3^	MFI
Choice	84.34 ^b^	59.25 ^b^	84.34 ^b^	59.25 ^b^
Select	79.20 ^b^	64.55 ^a^	79.20 ^b^	64.55 ^a^
Standard	105.70 ^a^	55.24 ^b^	105.70 ^a^	55.24 ^b^
Days ^2^				
1	118.42 ^a^	49.78 ^b^	118.42 ^a^	49.78 ^b^
3	95.31 ^b^	59.51 ^a^	95.31 ^b^	59.51 ^a^
7	86.70 ^bc^	62.02 ^a^	86.70 ^bc^	62.02 ^a^
14	73.55 ^c^	64.18 ^a^	73.55 ^c^	64.18 ^a^
21	74.76 ^c^	62.94 ^a^	74.76 ^c^	62.94 ^a^

^1^ Quality classification is all marbling categories within Choice, Select and Standard categories. ^2^ Postmortem days of aging of steaks at 4 °C in vacuum package. ^3^ Newtons can be translated to kg by dividing by 9.80665. ^a,b,c^ Different letters within a column are significantly different at *p* < 0.05.

**Table 4 animals-13-01947-t004:** Gene expression number of raw reads by treatment, quality of reads, and G and C content in reads.

		Adipose Tissue				
	Raw ^1^	Clean ^2^	Error (%)	Q20(%) ^3^	Q30(%) ^4^	GC Content (%) ^5^
Standard	52,994,544	51,102,307	0.036	96.59	91.53	54.05
Select	52,655,296	52,008,329	0.012	96.61	91.50	54.26
Choice	68,918,968	66,800,801	0.031	97.10	92.70	54.63
		Muscle Tissue				
	Raw ^1^	Clean ^2^	Error (%)	Q20(%) ^3^	Q30(%) ^4^	GC Content (%) ^5^
Standard	51,324,601	49,558,730	0.034	96.19	90.55	52.86
Select	57,170,688	55,292,644	0.033	96.46	91.12	54.16
Choice	48,957,502	46,709,699	0.046	96.62	91.41	54.79

^1^ Raw Reads: the original sequencing reads counts; ^2^ Clean Reads: number of reads after filtering; ^3^ Q20: percentages of bases whose correct base recognition rates are greater than 99% out of total bases; ^4^ Q30: percentages of bases whose correct base recognition rates are greater than 99.9% out of total bases; ^5^ GC content: percentages of G and C out of total bases.

**Table 5 animals-13-01947-t005:** Selected differentially expressed genes from adipose tissue from beef steers with different USDA carcass quality grades including gene name, gene Ensembl ID, calculated fold change, Bonferroni adjusted *p*-value, gene abbreviation.

**Standard to Select Adipose**				
Gene Code	Fold Change	*p*-adj	Gene Abbreviation	Gene Name
Downregulated:				
ENSBTAG00000008063	−1.59	2.47 × 10^−5^	PPARA	Peroxisome proliferator activated receptor alpha
ENSBTAG00000014387	−1.85	2.19 × 10^−5^	PRKAB2	Protein kinase AMP-activated non-catalytic subunit beta 2
ENSBTAG00000040128	−1.24	5.96 × 10^−5^	FZD4	Frizzled class receptor 4
ENSBTAG00000006037	−1.82	3.03 × 10^−5^	WISP2	WNT1 inducible signaling pathway protein 2
Upregulated:				
ENSBTAG00000021077	446	3.70 × 10^−8^	BOLA-DMB	Major histocompatibility complex, class II, DM beta
**Select to Choice Adipose**				
Gene Code	Fold Change	*p*-adj	Gene Abbreviation	Gene Name
Downregulated:				
ENSBTAG00000014911	3.65	2.46 × 10^−5^	LEP	Leptin
ENSBTAG00000018777	2.64	6.08 × 10^−7^	ADCY5	Adenylate cyclase type 5
**Standard to Choice Adipose**				
Gene Code	Fold Change	*p*-adj	Gene Abbreviation	Gene Name
Downregulated:				
ENSBTAG00000034222	1.18	4.76 × 10^−5^	CAB39L	Calcium binding protein 39 like
ENSBTAG00000005198	1.80	2.62 × 10^−7^	FGF1	Fibroblast growth factor 1
ENSBTAG00000047202	1.37	8.74 × 10^−6^	GRIN1	Glutamate ionotropic receptor NMDA type subunit 1
ENSBTAG00000014911	1.35	0.000297	LEP	Leptin
ENSBTAG00000013108	2.19	9.91 × 10^−5^	HK2	Hexokinase 2
ENSBTAG00000017567	1.79	1.48 × 10^−6^	ACC1	Acetyl-CoA carboxylase alpha
ENSBTAG00000045728	1.84	0.000115	SCD1	Stearoyl-CoA desaturase
ENSBTAG00000008102	2.34	3.52 × 10^−6^	CRTAC1	Cartilage acidic protein 1 isoform 2 precursor
ENSBTAG00000008153	1.03	0.032	CAMSAP2	Calmodulin regulated spectric associated protein family member 2
ENSBTAG00000011337	1.72	0.0037	ANKRD33B	Ankyrin repeat domain 33B
ENSBTAG00000013107	2.12	4.76 × 10^−6^	SHANK1	SH3 and multiple ankyrin repeat domains 1
ENSBTAG00000018473	3.70	0.039	MARCO	Macrophage recptor with collagenous structure
ENSBTAG00000026156	1.63	0.025	VCL	Vinculin
ENSBTAG00000015690	1.03	1.88 × 10^−6^	PLIN4	Perilipin 4
ENSBTAG00000003359	1.44	4.39 × 10^−6^	ELOVL5	ELOVL fatty acid elongase 5
Upregulated:				
ENSBTAG00000027654	−1.43	7.97 × 10^−5^	EIF4EBP1	Eukaryotic translation initiation factor 4E binding protein 1
ENSBTAG00000016071	−1.56	0.000195	HHIP	Hedgehog interacting protein
ENSBTAG00000003658	−1.23	4.50 × 10^−5^	RELN	Reelin precursor
ENSBTAG00000007446	−1.49	8.45 × 10^−5^	NGF	Nerve growth factor
ENSBTAG00000007446	−1.55	0.004	SCART1	Scavenger receptor family member expressed on T-cells
ENSBTAG00000007554	−1.26	0.022	IFI6	Interferon alpha inducible protein 6
ENSBTAG00000015182	−1.58	0.0002	STARD10	StAR related lipid transfer domain containing 10
ENSBTAG00000039520	−2.29	0.042	SIRPB1	Signal Regulatory Protein
ENSBTAG00000009656	−1.53	5.51 × 10^−6^	BOLA-DQA2	Major histocompatibility complex, class II, DQ alpha 2
ENSBTAG00000021077	−11.68	1.40 × 10^−6^	BOLA-DQB	Major histocompatibility complex, class II, DQ beta
ENSBTAG00000038128	−2.36	1.36 × 10^−6^	BOLA-DQA5	Major histocompatibility complex, class II, DQ alpha 5

**Table 6 animals-13-01947-t006:** Selected differentially expressed genes from muscle tissue from beef steers with different USDA carcass quality grades including gene name, gene Ensembl ID, calculated fold change, Bonferroni adjusted *p*-value, gene abbreviation.

**Standard to Choice Muscle**				
Gene Code	Fold Change	*p*-adj	Gene Abbreviation	Gene Name
Downregulated:				
ENSBTAG00000017412	−1.23	0.0.19	SOCS6	Suppressor of cytokine signaling 6
ENSBTAG00000021308	−1.12	0.032	IRS1	Insulin receptor substrate 1
Upregulated:				
ENSBTAG00000002362	1.69	0.0009	APOLD1	Apolipoprotein L domain containing 1
ENSBTAG00000032369	1.45	0.03	NMI	N-myc and STAT interactor
ENSBTAG00000009656	2.67	0.002	BOLA-DQA2	Major histocompatibility complex, class II, DQ alpha 2
ENSBTAG00000012451	1.36	0.041	BOLA-DMB	Major histocompatibility complex, class II, DM beta

**Table 7 animals-13-01947-t007:** GO enrichment of Select USDA quality grade compared to Standard USDA quality grade from beef adipose tissue including GO accession number, GO term description, GO category, corrected *p*-value, gene counts and up- or downregulation of counted genes.

Select Compared to Standard Adipose Tissues						
GO Accession	Description	Category	*p*-Adj	Count	Up	Down
GO:0006629	Lipid metabolic process	Cellular	0.042	956	4	0
GO:0010887	Negative regulation of cholesterol storage	Cellular	0.005	4	1	0
GO:0046426	Negative regulation of JAK-STAT cascade	Cellular	0.057	45	1	0
GO:0010888	Negative regulation of lipid storage	Cellular	0.017	12	1	0
GO:0010891	Negative regulation of sequestering of triglyceride	Cellular	0.007	5	1	0

**Table 8 animals-13-01947-t008:** GO enrichment of Select USDA quality grade compared to Choice USDA quality grade adipose tissue from beef cattle including GO accession number, GO term description, GO category, corrected *p*-value, gene counts and up- or downregulation of counted genes.

Select Compared to Choice Adipose Tissues						
GO Accession	Description	Category	*p*-Adj	Count	Up	Down
GO:0060612	Adipose tissue development	Cellular	0.031	28	0	1
GO:0046427	Positive regulation of JAK-STAT cascade	Cellular	0.045	48	0	1
GO:00045723	Positive regulation of fatty acid biosynthetic process	Cellular	0.013	14	0	1
GO:0045923	Postive regulation of fatty acid metabolic process	Cellular	0.026	26	0	1
GO:0046889	Positive regulation of lipid biosynthetic process	Cellular	0.043	44	0	1

**Table 9 animals-13-01947-t009:** GO enrichment of Choice USDA quality grade compared to Standard USDA quality grade adipose tissue from beef cattle including GO accession number, GO term description, GO category, corrected *p*-value, gene counts and up- or downregulation of counted genes.

Choice Compared to Standard Adipose Tissues						
GO Accession	Description	Category	*p*-Adj	Count	Up	Down
GO:0006633	Fatty acid biosynthetic process	Cellular	0.000	104	5	1
GO:0034625	Fatty acid elongation, monounsaturated fatty acid	Cellular	0.048	7	2	0
GO:0006631	Fatty acid metabolic process	Cellular	0.014	254	5	1
GO:0060612	Adipose tissue development	Cellular	0.024	28	2	0
GO:0045723	Positive regulation of fatty acid biosynthetic process	Cellular	0.004	14	2	0
GO:0045923	Postive regulation of fatty acid metabolic process	Cellular	0.016	26	2	0
GO:0004321	Fatty-acyl-CoA synthase activity	Cellular	0.033	4	0	1

**Table 10 animals-13-01947-t010:** GO enrichment of Standard USDA quality grade compared to Choice USDA quality grade muscle tissue from beef cattle including GO accession number, GO term description, GO category, corrected *p*-value, gene counts and up- or downregulation of counted genes.

Standard Compared to Choice Muscle Tissues						
GO Accession	Description	Category	*p*-Adj	Count	Up	Down
GO:0001578	Microtubule bundle formation	Cellular	0.006	75	1	1
GO:0046785	Microtubule polymerization	Cellular	0.073	51	0	1

## Data Availability

The data presented in this study are available on request from the corresponding author, including the FPKM of all genes, DEG outputs, and fold changes. The data are not publicly available because the raw sequencing data format is unsuitable for GEO submission.

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
