# Peer review of "Molecular Pathways for Muscle and Adipose Tissue Are Altered between Beef Steers Classed as Choice or Standard"

_animals, 2023, doi:10.3390/ani13121947_

Round 1
Reviewer 1 Report (New Reviewer)
The presented study is well performed and generated some interesting data. However, the description could be improved in some points as detailed below.
1. There is a section “3. Results” and a section “5. Conclusion”, but no “4. Discussion”.
2. The Materials and Methods as well as the Results should be structured in subchapters for more clarity.
3. Statistics need to consider repeated measurements in the model for aged samples.
4. Some discussion related to meat tenderness should be connected with transcriptome data. What can be concluded from this?
Minor items
L26 correct “…at harvestRNA was…”
L57-59 it should be mentioned that another breed was used in that study.
L78 correct “…average of 313 14 kg.”
L121-122 Clarify “Two samples per steer, per time point, yielded four measurements per steer.”, since you had 5 time points?
L128, 149, 150, 154 etc. and tables be consistent in use and format of P-values (p-value, P-value, P=, p=…)
L142, 147, 152-157 add the source or reference for the software and database used.
L174 and 187 clarify if the shear force, respectively the MFI, is calculated over all aging time points or what else the values represent.
L187 “Table 2” must be “Table 3”
L210-213 Needs clarification. It seems that in Figure 1 most values for muscle are >0.94?
L227, 228 etc. abbreviations must be explained at the first use. Official gene names and formats (italics) should be used always.
L227-229 a reference is needed
L252 it can be discussed in relation to the absence of a difference in muscle area among groups
Author Response
Reviewer #1
The presented study is well performed and generated some interesting data. However, the description could be improved in some points as detailed below.
Thank you for the thorough and thoughtful consideration of our manuscript.
- There is a section “3. Results” and a section “5. Conclusion”, but no “4. Discussion”.
We provided Results and Discussion together and have changed the section headings to show this.
- The Materials and Methods as well as the Results should be structured in subchapters for more clarity.
We have attempted to provide subheadings to improve clarity.
- Statistics need to consider repeated measurements in the model for aged samples.
We did not feel this was necessary because repeated measures implies there is a relationship between the samples. The variation of tenderness from front to back and side to side does not keep these as related. Further, the by statement for days of aging gives similar answers to the repeated measures. A statistician was consulted and he is the one that confirmed the similarity in the analysis.
- Some discussion related to meat tenderness should be connected with transcriptome data. What can be concluded from this?
We did not find changes in any genes known to be associated with meat tenderness. As was stated we feel that this is due to the muscle being relatively less controlled by transcriptional regulation at this time with the primary changes being seen in the adipose tissue. We have added text to the manuscript for clarity on this.
Minor items
L26 correct “…at harvestRNA was…” A space was added here.
L57-59 it should be mentioned that another breed was used in that study. This was clarified
L78 correct “…average of 313 14 kg.” The +/- was added.
L121-122 Clarify “Two samples per steer, per time point, yielded four measurements per steer.”, since you had 5 time points? This was clarified.
L128, 149, 150, 154 etc. and tables be consistent in use and format of P-values (p-value, P-value, P=, p=…) This has been made consistent across the text and tables.
L142, 147, 152-157 add the source or reference for the software and database used. These references have been added
L174 and 187 clarify if the shear force, respectively the MFI, is calculated over all aging time points or what else the values represent. This has been clarified
L187 “Table 2” must be “Table 3” Corrected
L210-213 Needs clarification. It seems that in Figure 1 most values for muscle are >0.94? This has been clarified.
L227, 228 etc. abbreviations must be explained at the first use. Official gene names and formats (italics) should be used always. The formatting has been corrected and the full names of each gene are listed.
L227-229 a reference is needed A reference was added.
L252 it can be discussed in relation to the absence of a difference in muscle area among groups. Discussion of this was added.
Reviewer 2 Report (New Reviewer)
This is an interesting manuscript aimed to study gene expression associated with metabolism and intercellular signaling from beef cattle muscular and adipose tissue. However, there are some procedures missing that are needed for an RNA-seq study. There are also several grammar suggestions to consider:
- Line 3: I suggest to use "beef steers” instead of “animals”.
- Lines 5-9: The same affiliation should not be repeated.
- Line 11: The word “correspondence” is repeated.
- Line 25: Replace “muscle adipose tissue” by “muscle and adipose tissue”.
- Lines 26-27: Replace the phrase “were collected at harvestRNA was harvested from tissues, and sequenced” by “were collected, and the RNA was harvested and sequenced”.
- Line 28: Replace “adjusted” by “were adjusted”.
- Line 35: How about genes detected from muscle tissue?
- Line 35: Which were the names of the enriched pathways’
- Line 36: Replace “varys” by “varies”.
- Lines 69-70: The objective should include gene and/or pathway analyses.
- Line 72: Insert an empty line.
- Line 78: It appears that a sign is missing between 313 and 14 kg.
- Line 134: I suggest describing briefly the RNA extraction procedure.
- Line 142: Indicate the complete name of the genome and its reference.
- Line 150: Clarify if fold change employed is equal to1 or higher than 1 (>1).
- Lines 146-157: Include the corresponding reference for each software used for analyses.
- Line 157: Describe the procedures used to construct the gene networks showed in figure 2.
- Line 157: Why the authors did not include a PCR gene validation? This procedure is required to confirm accuracy and reliability of RNAseq results.
- Line 242: I suggest to describe specific results showed in tables 7, 8 and 9.
- Line 270: Remove blue letters at the top and the bottom of the figure 2.
- Line 275: What is the difference between the title of table 7 and table 8. I suppose that one of them should talk about muscle.
- Discussion section appeared to be missing. Or Discussion was combined with Results? If so, please clarify.
- Line 297: The objective should not be included as conclusion.
- Line 325: Please correct all references following guidelines described in the “Instructions for Authors”.
Author Response
Reviewer #2
This is an interesting manuscript aimed to study gene expression associated with metabolism and intercellular signaling from beef cattle muscular and adipose tissue. However, there are some procedures missing that are needed for an RNA-seq study. There are also several grammar suggestions to consider:
- Line 3: I suggest to use "beef steers” instead of “animals”. This has been changed
- Lines 5-9: The same affiliation should not be repeated. This was done according to the template provided.
- Line 11: The word “correspondence” is repeated. This has been corrected.
- Line 25: Replace “muscle adipose tissue” by “muscle and adipose tissue”. This has been corrected.
- Lines 26-27: Replace the phrase “were collected at harvestRNA was harvested from tissues, and sequenced” by “were collected, and the RNA was harvested and sequenced”. This has been corrected.
- Line 28: Replace “adjusted” by “were adjusted”. This has been corrected.
- Line 35: How about genes detected from muscle tissue? This is in the previous sentence.
- Line 35: Which were the names of the enriched pathways’ This was limited by word count in the abstract but is included in the results and discussion section.
- Line 36: Replace “varys” by “varies”. This has been corrected.
- Lines 69-70: The objective should include gene and/or pathway analyses. This has been added.
- Line 72: Insert an empty line. This has been done.
- Line 78: It appears that a sign is missing between 313 and 14 kg.
- Line 134: I suggest describing briefly the RNA extraction procedure. A brief description of the extraction was added.
- Line 142: Indicate the complete name of the genome and its reference. This information has been added.
- Line 150: Clarify if fold change employed is equal to1 or higher than 1 (>1). This has been clarified.
- Lines 146-157: Include the corresponding reference for each software used for analyses. These have been added.
- Line 157: Describe the procedures used to construct the gene networks showed in figure 2.
This has been added.
- Line 157: Why the authors did not include a PCR gene validation? This procedure is required to confirm accuracy and reliability of RNAseq results. This topic has been quite controversial for some time but the current consensus is that RNAseq is more accurate and repeatable than qPCR methods and that qPCR validation is not necessary in most cases. As we are not focusing on rare transcripts, or variants the literature indicates that there is no benefit to qPCR validation.
A comprehensive assessment of RNA-seq accuracy, reproducibility and information content by the Sequencing Quality Control Consortium." Nature biotechnology 32, no. 9 (2014): 903-914.
Wang, Liguo, Shengqin Wang, and Wei Li. "RSeQC: quality control of RNA-seq experiments." Bioinformatics 28, no. 16 (2012): 2184-2185.
Chen, Wanqiu, Yongmei Zhao, Xin Chen, Zhaowei Yang, Xiaojiang Xu, Yingtao Bi, Vicky Chen et al. "A multicenter study benchmarking single-cell RNA sequencing technologies using reference samples." Nature Biotechnology 39, no. 9 (2021): 1103-1114.
't Hoen, Peter AC, Marc R. Friedländer, Jonas Almlöf, Michael Sammeth, Irina Pulyakhina, Seyed Yahya Anvar, Jeroen FJ Laros, Henk PJ Buermans, Olof Karlberg, and Mathias Brännvall. "Reproducibility of high-throughput mRNA and small RNA sequencing across laboratories." Nature biotechnology 31, no. 11 (2013): 1015-1022.
Williams, Alexander G., Sean Thomas, Stacia K. Wyman, and Alisha K. Holloway. "RNA‐seq data: challenges in and recommendations for experimental design and analysis." Current protocols in human genetics 83, no. 1 (2014): 11-13.
- Line 242: I suggest to describe specific results showed in tables 7, 8 and 9. These results have now been described in the text.
- Line 270: Remove blue letters at the top and the bottom of the figure 2. This has been corrected.
- Line 275: What is the difference between the title of table 7 and table 8. I suppose that one of them should talk about muscle. Table 7 compares Select to Standard Adipose; Table 8 compares Select to Choice Adipose
- Discussion section appeared to be missing. Or Discussion was combined with Results? If so, please clarify. It was combined and this has been clarified.
- Line 297: The objective should not be included as conclusion. This has been reworded for clarity.
- Line 325: Please correct all references following guidelines described in the “Instructions for Authors”. Reference format has been corrected per guidelines.
Round 2
Reviewer 2 Report (New Reviewer)
The manuscript has improved significantly. However, there are still some minor grammar comments that should be considered:
- Line 11: A round bracket appears to be missing.
- Line 29: Please verify the criteria for log2. It usually should be >1.
- Line 29: Replace “was” by “were”.
- Line 70: Please remove the underlined.
- Line 148: Replace “Bos taurus” by “Bos taurus”.
- Line 156: Please verify if fold change < 1 is the right criteria. It usually should be > 1 (i.e., greater than 1).
- Lines 158-164: The paragraph should not be in bold style.
- Line 166: Move “3. Results and Discussion” to the next line.
- Lines 256-263: Please remove the underlined.
- Lines 274-276: Please remove the underlined.
- Line 316: Replace “tissue” by “tissues”.
- Line 321: Replace “shows” by “showed”.
- Lines 332-334: Please remove the underlined.
- Line 39: In References section, several references should be corrected following the “Instructions for authors” guidelines (i.e., Author 1, A.B.; Author 2, C.D. Title of the article. Abbreviated Journal Name Year, Volume, page range).
Author Response
Thank you for the great comments and your assistance in improving our manuscript.
Line 11: A round bracket appears to be missing. -Corrrected
- Line 29: Please verify the criteria for log2. It usually should be >1. -Corrrected
- Line 29: Replace “was” by “were”. -Corrrected
- Line 70: Please remove the underlined. -Corrrected
- Line 148: Replace “Bos taurus” by “Bos taurus”. -Corrrected
- Line 156: Please verify if fold change < 1 is the right criteria. It usually should be > 1 (i.e., greater than 1). -Corrrected
- Lines 158-164: The paragraph should not be in bold style. -Corrrected
- Line 166: Move “3. Results and Discussion” to the next line. -Corrrected
- Lines 256-263: Please remove the underlined. -Corrrected
- Lines 274-276: Please remove the underlined. -Corrrected
- Line 316: Replace “tissue” by “tissues”. -Corrrected in all of the similar tables
- Line 321: Replace “shows” by “showed”. -Corrrected
- Lines 332-334: Please remove the underlined. -Corrrected
- Line 39: In References section, several references should be corrected following the “Instructions for authors” guidelines (i.e., Author 1, A.B.; Author 2, C.D. Title of the article. Abbreviated Journal Name Year, Volume, page range). We have endeavored to get them all into the desired format.
This manuscript is a resubmission of an earlier submission. The following is a list of the peer review reports and author responses from that submission.
Round 1
Reviewer 1 Report
This manuscript is well organized and is clear in intent, hypothesis and conclusions. There are many areas that need improvement prior to publication.
Materials and Methods
Line 88 How long were steers on feed for each treatment? Were they all slaughtered on the same day?
Line 106 For collection of MFI samples from one steak which were aged as smaller samples: was location of each sample from the steak randomized for day of aging as MFI is not consistent from medial to lateral locations within strip steaks? As sample size is small, this would be important in the explanation of the MFI value for select vs choice and standard.
Line 124 Were two samples per steer also per aging time?
Line 132 What was the time points after stun of collection of subcutaneous fat, intramuscular fat, and muscle?
Line 143 Table 4
Line 175 Table 3
Table 2. Need to include standard error or other indicator of variance for each mean or group.
Table 3 seems to be missing information as there is no treatments listed and no superscripts that are shown which are indicated in the footnotes.
Table 4 need superscripts and more description of each column.
Figure 1 is inadequate as it cannot be read either as printed or from the pdf. The figure description needs to include what factor was used for the correlation. Is it quantity?
There needs to be further discussion on the pathways that are found to be part of the differences in gene expression, particularly the inflammatory indicators. There is a lot of interest in this area of growth and the discussion needs to be expanded to help explain why this would be an important factor.
Tables 5-10 Table titles need to be more descriptive as they should be able to be read without having to find the information in the text of the manuscript.
Reviewer 2 Report
Main concern regarding this manuscript is very high percentage of matching with somekind of report under same title available at https://www.montana.edu/extension/coa/documents/2018reports/Thomson1.pdf. Please elaborate this. Other comments are in attachtment.
